# LowDiff: Efficient Diffusion Sampling with Low-Resolution Condition

## Abstract

Diffusion models have achieved remarkable success in image generation but their practical application is often hindered by the slow sampling speed. Prior efforts of improving efficiency primarily focus on compressing models or reducing the total number of denoising steps, largely neglecting the possibility to leverage multiple input resolutions in the generation process. In this work, we propose LowDiff, a novel and efficient diffusion framework based on a cascaded approach by generating increasingly higher resolution outputs. Besides, LowDiff employs a unified model to progressively refine images from low resolution to the desired resolution. With the proposed architecture design and generation techniques, we achieve comparable or even superior performance with much fewer high-resolution sampling steps. LowDiff is applicable to diffusion models in both pixel space and latent space. Extensive experiments on both conditional and unconditional generation tasks across CIFAR-10, FFHQ and ImageNet demonstrate the effectiveness and generality of our method. Results show over 50% throughput improvement across all datasets and settings while maintaining comparable or better quality. On unconditional CIFAR-10, LowDiff achieves an FID of 2.11 and IS of 9.87, while on conditional CIFAR-10, an FID of 1.94 and IS of 10.03. On FFHQ 64×64, LowDiff achieves an FID of 2.43, and on ImageNet 256×256, LowDiff built on LightningDiT-B/1 produces high-quality samples with a FID of 4.00 and an IS of 195.06, together with substantial efficiency gains.

## 1 Introduction

High-fidelity image generation has become a reality thanks to powerful diffusion models, which have set new benchmarks across a wide range of generative tasks Baldridge et al. (2024); Betker et al. (2023); Brooks et al. (2024); Esser et al. (2024); Karras et al. (2024) Although early research mainly focuses on improving generation quality, there is a growing emphasis on accelerating the sampling process to make diffusion models more practical for real-world applications, especially in scenarios with limited computational resources or strict efficiency requirements Li et al. (2024a; 2023b;c); Shang et al. (2023); Sui et al. (2024); Wu et al. (2024). As image generation scales to higher resolutions and larger datasets, the demand for efficient training and inference has become increasingly critical.

To enable efficient diffusion models, previous work has primarily explored three strategies: model compression, denoising step reduction, and lower-dimension generation. First, various ***model compression*** techniques, including network pruning Fang et al. (2023); Wan et al. (2025), efficient architectural design Yang et al. (2023); Zhu et al. (2024); Phung et al. (2023); Li et al. (2023a), neural architecture search Tang et al. (2023); Li et al. (2023c), and quantization Sui et al. (2024); Li et al. (2023b), have been investigated to reduce the size of diffusion models and the latency per function evaluation. Second, ***denoising step reduction*** targets the number of function evaluations (NFEs), which can range from tens to thousands and is a major bottleneck in sampling efficiency. By incorporating techniques including knowledge distillation Luhman & Luhman (2021); Zhang & Ma (2024), progressive distillation Salimans & Ho (2022); Meng et al. (2023); Li et al. (2023c), and auxiliary networks or loss functions Wang et al. (2022); Xiao et al. (2021); Yin et al. (2024b;a), diffusion models can be accelerated while maintaining generation quality, often by allowing larger denoising step size.

The third approach, ***lower-dimension generation***, follows a divide-and-conquer strategy by breaking down the complex generation task into simpler sub-tasks. For instance, Stable Diffusion Rombach et al. (2022) integrates a variational autoencoder (VAE) and a diffusion model by applying the diffusion process in latent space and decoding the resulting latent variables into high-quality images. Recent work Li et al. (2024b) also uses intermediate features generated by diffusion models as conditional inputs to enhance subsequent stages of generation. Beyond improving efficiency, some other works also adopt similar hierarchical ideas to improve the quality of generated images Ho et al. (2022); Atzmon et al. (2024); Gu et al. (2023). However, the potential of leveraging low-resolution generation within these frameworks to improve sampling efficiency remains underexplored.

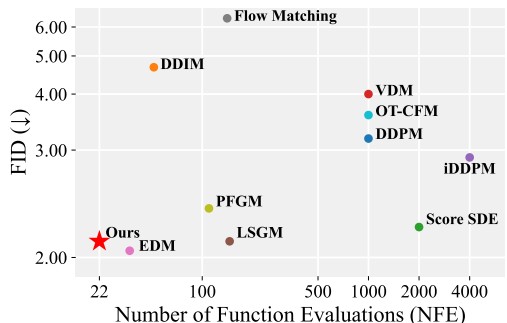

Figure 1: Sample quality vs. effective number of function evalutions. We compare different diffusion models that are trained from scratch on CIFAR10 dataset, and measure the generation quality by FID (↓). Our model provides comparable output quality as the previous best results and reduce the required computing efforts by more than 30%.

From the perspective of low-resolution generation, we investigate an alternative approach towards efficient diffusion models (i.e., faster sampling and lower memory footprint without sacrificing generation quality). While cascaded diffusion frameworks have been explored previously Ho et al. (2022); Li et al. (2024b), existing works primarily focus on improving perceptual quality and typically operate at relatively high starting resolutions (e.g., $32\times32$ or $64\times64$). Moreover, these methods commonly train separate diffusion models for each resolution stage, resulting in substantial memory overhead and slow sampling. To address these limitations, we propose LowDiff, a novel and efficient diffusion framework that leverages a cascaded generation strategy and is applicable to both UNet-based and Transformer-based architectures. To improve the sampling speed, LowDiff departs from previous designs and begins generation from an extremely low resolution (e.g., $8\times8$), which substantially reduces the number of denoising steps needed at higher resolutions (e.g., $64\times64$), requiring only half as many or even fewer steps. Since diffusion model complexity scales quadratically with spatial resolution, operating more of the sampling trajectory at ultra-low resolutions provides immediate efficiency gains. To avoid the memory footprint and training cost of maintaining multiple independent models, we introduce a unified diffusion architecture in which models across different resolutions share the vast majority of their parameters. Only lightweight, resolution-specific input/output adapters and resolution-aware embedding layers are introduced. This unified design is motivated by the structure of UNet models, whose stacked downsampling and upsampling layers naturally operate over multiple spatial resolutions (e.g., $8\times8$), suggesting that multiple resolutions can be operated within a single model without architectural redundancy. In addition, recent transformer-based diffusion models such as DiffusionTransformer Peebles & Xie (2023) operate in latent spaces whose resolutions (e.g., $8\times8$ and $16\times16$) fall squarely within the low-resolution regime. We therefore also evaluate LowDiff on both UNet-based and DiT-based diffusion models, confirming that the unified multi-resolution design generalizes beyond convolutional architectures.

Leveraging both (i) super-low-resolution cascaded generation and (ii) a unified weight-sharing diffusion backbone, LowDiff achieves faster sampling and lower memory usage, while maintaining comparable or even improved image quality compared with the baselines. Importantly, unlike distillation-based acceleration techniques, our method is trained end-to-end from scratch, without auxiliary models or loss terms, simplifying the training pipeline and improving stability. Our contribution is summarized as the following:

- **End-to-End Unified Multi-Resolution Cascaded Diffusion Framework.** We introduce a unified diffusion framework that performs cascaded generation across multiple resolutions within a single model, starting from ultra-low-resolution images to achieve efficient sampling.

- **Architecture-Agnostic Design.** Our framework is compatible with both UNet-based and Transformer-based diffusion architectures, enabling broad applicability across different diffusion backbones.

- **Improved Generation Quality-Efficiency Trade-off.** Extensive experiments show that our method delivers comparable or superior image quality across multiple datasets while improving throughput by more than 50% compared to standard baselines.

## 2 RELATED WORKS

**Efficient Diffusion Models.** Diffusion models are typically bulky in size and require a large amount of denoising steps to generate one output. To reduce the size and memory requirement of diffusion models, a structural pruning technique is proposed by identifying important weights according to the informative gradients Fang et al. (2023). A progressive soft pruning strategy is investigated based on the gradient flow of the energy function Wan et al. (2025). To quantize diffusion models, Q-Diffusion Li et al. (2023b) applies post-training quantization techniques with timestep-aware calibration and shortcut quantization split to shrink the model weights to 4-bit. SVDQuant Li et al. (2024a) quantizes both weights and activations to 4-bit by outlier redistribution. BitsFusion Sui et al. (2024) further reduces the model size below 2-bit by weight quantization-aware training, where during training, time steps are sampled according to the corresponding quantization error of each timestep. To reduce the number of function evaluations (NFEs), distillation technique to progressively reduce the required number of denoising steps is proposed in Salimans & Ho (2022). SnapFusion Li et al. (2023c) further proposes a CFG-aware step distillation method to take into account the effect of classifier-free guidance, thereby improves the generate quality. Adversarial loss is also introduced to combine the advantages of diffusion models and GAN models in Xiao et al. (2021); Wang et al. (2022). Meanwhile, the serial studies of DMD Yin et al. (2024b;a) propose to use distribution matching distillation to help reduce the required denoising steps. These efficiency-oriented techniques primarily focus on compressing the U-Net architecture or reducing the number of denoising steps, rather than leveraging low-resolution generation to accelerate the overall process.

**MultiScale Diffusion Models.** Multiscale generation with diffusion models aim to improve the generation performance by operating across multiple resolutions or scales, which are conventionally studied in generating high-resolution images. Such high-resolution generators typically generate images with a resolution of at least 256 and start from 32 or 64 as the lowest resolution. Rather than generating high-resolution outputs directly from noise, these models decompose the generation into sequential stages, starting from a coarse representation and progressively refining finer details. The cascaded approach is one of the most common methods to implement multiscale generation Balaji et al. (2022); Ho et al. (2022); Ramesh et al. (2022); Saharia et al. (2022a;b). While effective, these methods rely on training a series of separate and independent models, each responsible for one image resolution, along with thousands of denoising steps for generation. This introduces substantial memory overhead, storage costs, and operational complexity during inference. Another line of work explores pyramidal frameworks, using outputs of different scales without any cascaded structure to improve the generation quality Atzmon et al. (2024); Gu et al. (2022); Jin et al. (2024); Teng et al. (2023). These methods usually work on the lower resolution images/videos first and then continue to denoising the higher resolution counterpart with modification on the image signal (e.g., noise level). Although, these methods all show improvements but only on high-resolution datasets and these complex analysis and modifications are required to obtain satisfied performance. Matryoshka Diffusion Model (MDM) Gu et al. (2023) propose to use a unified architecture to help with generation by integrating with lower resolution image generation. This design unifies the multiscale generation process in a single model and avoids distribution mismatches, which promotes model consistency and simpler inference compared with CDMs. However, these models usually rely on increased architectural complexity and have issues related to training stability and efficiency. Thus, these previous models are not purposed to implement efficient diffusion models.

## 3 METHODS

In this section, we introduce our method for efficient diffusion models. Our approach can be naturally implemented using a Transformer-based diffusion model because the Transformer-based architectures are inherently resolution-flexible since they operate on image patches which makes that a unified Transformer can process multiple resolutions simply by adding resolution embeddings. This makes the implementation straightforward, as the patch-based design avoids the need for specialized

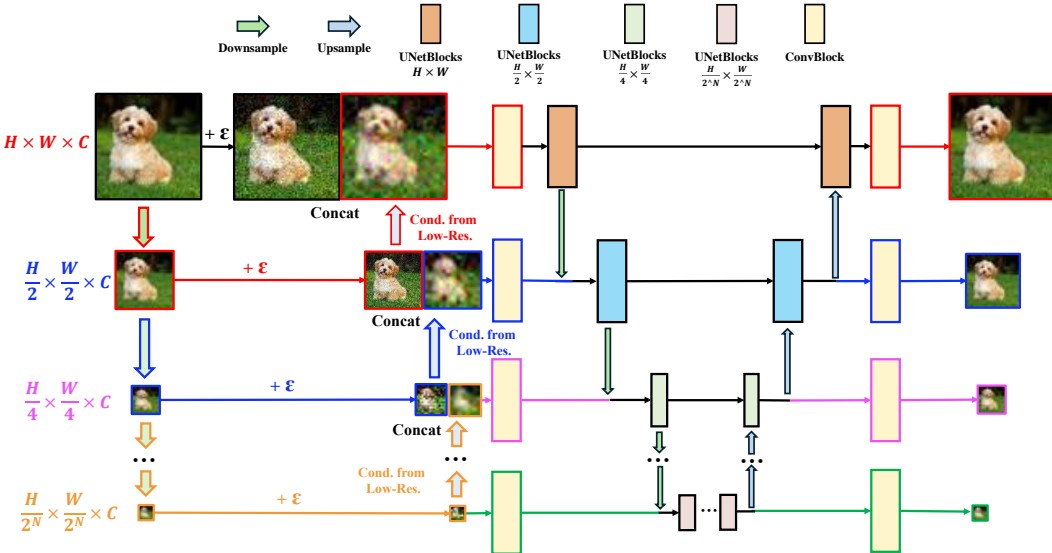

Figure 2: **Model Architecture.** Our model supports progressive training at arbitrary resolution levels. During the training process, the training for the lowest resolution images is standard and unconditional and the training for higher resolution images is conditioned on the upsampled corresponding lower-resolution noisy images (injected with independent Gaussian noise).

resolution-dependent components. In contrast, implementing U-Nets to support a unified multi-resolution pipeline is less direct, as convolutional structures are tightly coupled with resolution. Therefore, we present our method with U-Net as the core example, highlighting the architectural adaptations required to achieve efficiency. The core idea is to first generate a low-resolution image that captures the coarse structure of the target, and then progressively refine it through a series of cascaded diffusion stages. To enhance robustness across stages, we adopt truncation augmentation, where partially denoised low-resolution outputs are used as conditioning inputs for subsequent stages. To further improve efficiency, we propose a unified U-Net architecture shared across all resolutions. Instead of training separate models, we share the core network blocks and introduce lightweight, resolution-specific input/output convolutional layers, along with resolution-aware embeddings, to differentiate between resolutions while minimizing architectural overhead.

### 3.1 LOW-RESOLUTION CONDITIONING FOR EFFICIENT SAMPLING

Diffusion-based image generation suffers from significant computational overhead when targeting high-resolution outputs, as both memory usage and computation scale quadratically with image resolution. Consequently, performing a large number of denoising steps directly at high resolutions often leads to inefficiencies that severely limit scalability. Although reducing the number of steps can speed up sampling, it typically comes at the cost of degraded generation quality. To address this trade-off, we hypothesize that the early stages of the generation process do not require high-resolution outputs and can instead be performed more efficiently at lower resolutions. Specifically, we first generate a low-resolution output, which is simpler and requires much reduced computational cost. After that, we generate the high-resolution images by conditioning on the generated low-resolution outputs, allowing us to reduce the number of denoising steps required for this high-resolution generation. Such a conditional generation approach enables efficient generation while maintaining high-quality results.

### 3.2 CASCADED DIFFUSION GENERATION

To further enhance generation efficiency, we extend the above approach into a hierarchical framework, where each stage conditions the generation on the low-resolution output produced in the preceding stage, and the entire pipeline begins with an unconditional generation at the lowest resolution. For this purpose, we follow the previous techniques proposed in CDM Ho et al. (2022) to train the diffusion model, Specifically, we adopt the truncation augmentation technique to ensure robust

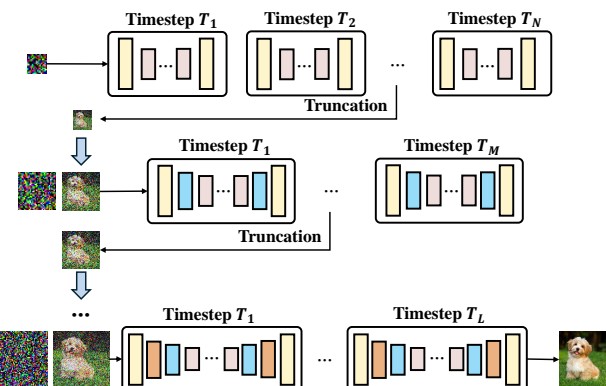

Figure 3: **Inference Pipeline.** The generation proceeds in a bottom-up manner: starting from Gaussian noise, the model first generates the lowest resolution image with truncation. This noisy image is then upsampled to help with higher resolution image generation, which in turn is used to condition the next-resolution output until generating the highest resolution image.

generation. For this purpose, during training, as illustrated in Fig. 2, instead of conditioning high-resolution generation on a fully denoised low-resolution image, we sample an independent noise $\varepsilon_2$ for the low-resolution image and obtain a noisy variant $x^{\text{low}}_{\sigma_2}$. This partially denoised image is then upsampled and concatenated with the high-resolution noisy input $x^{\text{high}}_{\sigma_1}$ to feed into the high-resolution denoising network. During sampling, as shown in Fig. 3, the lowest-resolution image is generated unconditionally using the standard denoising process from pure noise. Except for the final desired resolution, this denoising process is truncated at an intermediate noise level, and the partially denoised output is upsampled to condition the following high-resolution generation. This conditional generation process is repeated until the final clean image of the desired resolution is obtained.

### 3.3 NETWORK REUSING AND UNIFIED ARCHITECTURE FOR MULTI-RESOLUTION GENERATION

Unlike previous methods Ho et al. (2022); Saharia et al. (2022b) that train a separate diffusion model for each resolution, we further improve efficiency by reusing blocks within a single U-Net architecture. Leveraging the self-similarity property of U-Net, we repurpose its internal low-resolution sub-network to serve dual roles, as illustrated in Fig. 4. Specifically, for low-resolution image generation, the sub-network functions as the primary components of the U-Net. In contrast, for higher-resolution generation, the same blocks are used to process downsampled features, seamlessly integrating into the larger network hierarchy. Besides, we introduce two additional light-weight components to the original U-Net, as explained in the following. The final U-Net structured with the proposed I/O and embedding layers is shown in Fig. 2.

**Resolution-specific I/O convolutions.** Since the low-resolution sub-network are reused for both low- and high-resolution generation, we introduce resolution-specific input and output convolutional layers to mitigate potential distributional discrepancies between the two usages. Specifically, for each supported resolution level (i.e., $H \times W$, $\frac{H}{2} \times \frac{W}{2}$, ..., $\frac{H}{2^N} \times \frac{W}{2^N}$), we add independent convolutional layers at the input and output to align feature dimensions associated with different resolutions. This modular design enables the model to accept and generate images at varying resolutions, which is essential for supporting resolution-conditional training within a unified architecture.

**Resolution embedding.** To further improve the flexibility of the reused low-resolution sub-network, we introduce resolution-information into these blocks to make them resolution-aware during functioning. Specifically, we encode the resolution label using a one-hot vector, which is transformed through a learnable mapping layer. The resulting resolution embedding is added to other embeddings (e.g., noise embedding, label embedding, etc.), and the combined embedding will be used for training the network. This design allows the model to distinguish features generated at different resolutions and encourages shared layers to learn resolution-specific behaviors.

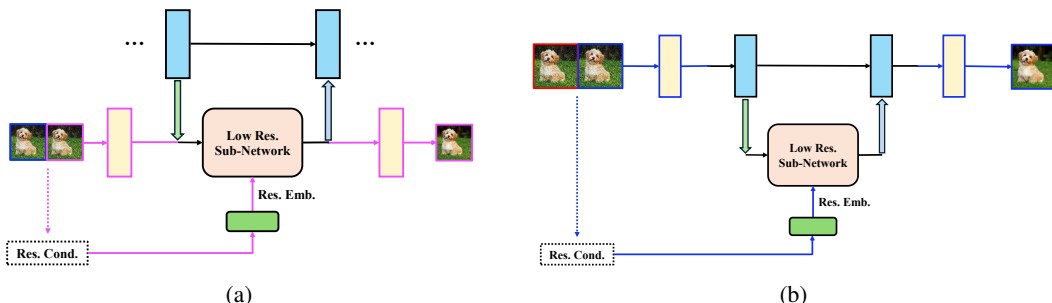

(a)                                                           (b)

Figure 4: **Dual usage of low-resolution blocks in our architecture.** "Res. Cond." denotes the target resolution (e.g., 8, 16). (a) Low-resolution image generation. The low-resolution sub-network operates independently to generate low-resolution images, taking only the input low-resolution image feature and the corresponding low-resolution embedding as inputs. The pink arrows indicate the forward direction of this process. (b) Sub-network reuse for high-resolution generation. The same low-resolution sub-network is reused as part of the high-resolution pathway, where it serves as an internal module that processes downsampled high-resolution features together with the high-resolution embedding. The blue arrows indicate the forward direction in this high-resolution generation process. Instead of operating as an independent module, they are integrated into the high-res network to follow a UNet-style hierarchical design.

### 3.4    TRAINING AND SAMPLING WITH THE UNIFIED ARCHITECTURE

To clarify how to utilize the unified architecture we proposed above, here we elaborate on the training and sampling procedure with our model. As explained above, the training process can be divided into two categories: unconditional training on lowest resolution and low-resolution conditioned training for subsequent resolutions. Specifically, suppose that we have a predefined resolution set res $= \{r_1, r_2, ..., r_N\}$ and $r_1 > ... > r_N$. For the lowest-resolution image $x_0^{r_N}$, i.e., the one of $\frac{H}{2^N} \times \frac{W}{2^N}$ in Fig. 2, we apply unconditional diffusion generation with the corresponding blocks, i.e., the bottom row of green I/O layers with the innermost blocks of the original U-Net. Thus, the loss for this resolution is calculated by only activating the associated blocks, i.e. the green blocks in Fig. 2. Then, for each higher resolution, say the one of $\frac{H}{2} \times \frac{W}{2}$ in Fig. 2, we apply conditional diffusion generation. Specifically, we first down-sample the clean high-resolution image, which is the one of $\frac{H}{2} \times \frac{W}{2}$ here, to a lower resolution $\frac{H}{4} \times \frac{W}{4}$, and perturb them with Gaussian noise $\varepsilon_c$, which is independently distributed from the noise for perturbing the high-resolution input image of $\frac{H}{2} \times \frac{W}{2}$. This noisy low-resolution conditional image is then upsampled back to the original higher resolution $\frac{H}{2} \times \frac{W}{2}$. This upsampled image is concatenated with the noisy high-resolution image of $\frac{H}{2} \times \frac{W}{2}$ and fed into the network. To calculate the loss for this resolution, again, only blocks related to this resolution are activated, in this case the blue I/O layers with the sub-network of the U-Net with input resolution lower than $\frac{H}{2} \times \frac{W}{2}$. The above progress is repeated for each resolution and all losses are added together to get the final loss to update the network. The whole training algorithm is summarized in Algorithm 1.

For generation, as illustrated in Fig. 3, the process follows a bottom-up manner, where we start the generation from the lowest resolution. Specifically, we first sample a Gaussian noise at that resolution and follow the denoising procedure in standard diffusion models. Besides, during this step, only the sub-network processing the lowest resolution is activated, together with the corresponding I/O and embedding layers. Also, as mentioned above, the denoising process is truncated, i.e., stopped at some intermediate noise level, and the generated noisy output is upsampled to the next resolution. For each subsequent resolution, we use the upsampled noisy output generated in the preceding stage as a condition, and concatenate it with the noisy input of current resolution to feed into the model for the denoising purpose. Again, each step only involves the corresponding sub-network together with the I/O and embedding layers. This procedure is repeated in progressively increasing resolution, until the final desired resolution where we do not apply truncation but continue the

Table 1: Performance comparison of unconditional generation on CIFAR10 Dataset

| Model | NFE | FID $\downarrow$ | IS $\uparrow$ |
|---|---|---|---|
| DDPM Ho et al. (2020) | 1000 | 3.17 | 9.46 |
| DDIM Song et al. (2020a) | 50 | 4.67 | - |
| Score SDE Song et al. (2020b) | 2000 | 2.20 | 9.89 |
| VDM Kingma et al. (2021) | 1000 | 4.0 | - |
| iDDPM Nichol & Dhariwal (2021) | 4000 | 2.9 | - |
| LSGM Vahdat et al. (2021) | 147 | 2.10 | - |
| EDM Karras et al. (2022) | 35 | 2.04 Song et al. (2023) | 9.84 |
| Flow Matching Lipman et al. (2022) | 142 | 6.35 | - |
| PFGM Xu et al. (2022) | 110 | 2.35 | 9.68 |
| OT-CFM Tong et al. (2023) | 1000 | 3.57 | - |
| **Ours** | 22† | 2.11 | 9.87 |

†Effective NFE at a resolution of $32 \times 32$ obtained by converting the low-resolution NFEs using a latency-based scaling factor.

Table 2: Performance comparison of generation with classifier-free guidance on ImageNet256×256

| Model | Params | Epochs | NFE | FID$\downarrow$ | sFID$\downarrow$ | IS$\uparrow$ | Pre.$\uparrow$ | Rec.$\uparrow$ |
|---|---|---|---|---|---|---|---|---|
| LDM Rombach et al. (2022) | 400M | 200 | 250 | 3.60 | - | 247.7 | 0.87 | 0.48 |
| DiT-XL/2 Peebles & Xie (2023) | 675M | 1400 | 250 | 2.27 | 4.60 | 278.2 | 0.83 | 0.57 |
| SiT-XL/2 Ma et al. (2024) | 675M | 1400 | 250 | 2.06 | 4.50 | 270.3 | 0.82 | 0.59 |
| REPA + SiT-XL/2 Yu et al. (2024) | 675M | 800 | 250 | 1.42 | 4.70 | 305.7 | 0.80 | 0.65 |
| RelayDiffusion Teng et al. (2023) | - | - | 250 | 1.87 | 3.97 | 278.75 | 0.81 | 0.59 |
| PixelFlow Chen et al. (2025) | 677M | 320 | 250 | 1.98 | 5.83 | 282.1 | 0.81 | 0.60 |
| PixelNerd Wang et al. (2025a) | 700M | 160 | 250 | 2.15 | 4.55 | 297 | 0.79 | 0.59 |
| DDT-XL Wang et al. (2025b) | 675M | 400 | 250 | 1.26 | - | 310.6 | 0.79 | 0.65 |
| LightningDiT-XL Yao et al. (2025) | 675M | 800 | 250 | 1.35 | 4.15 | 295.3 | 0.79 | 0.65 |
| **Ours** | 677M | 320 | 175† | 1.79 | 4.26 | 225.8 | 0.78 | 0.64 |

†Effective NFE at a resolution of $16 \times 16$ (latent space) obtained by converting the low-resolution $(8 \times 8)$ (latent space) NFEs using a latency-based scaling factor.

denoising to get the final clean image. The sampling algorithm is summarized in Algorithm 2.

---

**Algorithm 1** Training

1: **repeat**
2:     **for** $i = N$ to 1 **do**
3:         $\boldsymbol{x}_0 \sim q(\boldsymbol{x}_0)$, $\boldsymbol{\varepsilon} \sim \mathcal{N}(\mathbf{0}, \sigma^2 \mathbf{I}^{\mathrm{r}_i \times \mathrm{r}_i})$
4:         $\boldsymbol{x}_0^{\mathrm{r}_i} \leftarrow \text{DOWNSAMPLE}(\boldsymbol{x}_0, \mathrm{r}_i)$
5:         **if** $i == N$ **then**
6:             $\boldsymbol{x}_c \leftarrow \varnothing$
7:         **else**
8:             $\boldsymbol{x}_c \leftarrow \text{DOWNSAMPLE}(\boldsymbol{x}_0, \mathrm{r}_{i+1})$
9:             $\boldsymbol{\varepsilon}_c \sim \mathcal{N}(\mathbf{0}, \sigma^2 \mathbf{I}^{\mathrm{r}_{i+1} \times \mathrm{r}_{i+1}})$
10:        $\boldsymbol{x}_c \leftarrow \text{UPSAMPLE}(\boldsymbol{x}_c + \boldsymbol{\varepsilon}_c, \mathrm{r}_i)$
11:        **end if**
12:        Backward with $\nabla_\theta \mathcal{L}(\hat{\boldsymbol{x}}_\theta^{\mathrm{r}_i}; \boldsymbol{x}_0^{\mathrm{r}_i}, \boldsymbol{\varepsilon}, \mathrm{r}_i)$
13:     **end for**
14: **until** converged

**Algorithm 2** Sampling

1: **for** $i = N$ to 1 **do**
2:     **if** $i == N$ **then**
3:         $\boldsymbol{x}_c \leftarrow \varnothing$
4:     **else**
5:         $\boldsymbol{x}_c \leftarrow \text{UPSAMPLE}(\boldsymbol{x}^{\mathrm{r}_{i+1}}, \mathrm{r}_i)$
6:     **end if**
7:     $\boldsymbol{x}_T^{\mathrm{r}_i} \sim \mathcal{N}(\mathbf{0}, \mathbf{I}^{\mathrm{r}_i \times \mathrm{r}_i})$
8:     **for** $t = T^{(\mathrm{r}_i)}$ to $\widetilde{T}^{(\mathrm{r}_i)}$ **do**
9:         $\boldsymbol{x}_{t-1}^{\mathrm{r}_i} \leftarrow \hat{\boldsymbol{x}}_\theta(\boldsymbol{x}_t^{\mathrm{r}_i}, \boldsymbol{x}_c, \sigma_t, \mathrm{r}_i)$
10:     **end for**
11:     $\boldsymbol{x}^{\mathrm{r}_i} \leftarrow \boldsymbol{x}_{\widetilde{T}^{(\mathrm{r}_i)}-1}^{\mathrm{r}_i}$
12: **end for**
13: $\boldsymbol{x}_0 \leftarrow \boldsymbol{x}^{\mathrm{r}_i}$
14: **return** $\boldsymbol{x}_0$

---

## 4 EXPERIMENTS

### 4.1 EXPERIMENTAL SETUP

Our model is designed to be compatible with any diffusion variant. To demonstrate this generality, we test our framework in both pixel space and latent space. For fair and consistent evaluation, we adopt the Variance Preserving (VP) diffusion framework as implemented in EDM Karras et al. (2022) [1] and the DiT framework as implemented in LightningDiT Yao et al. (2025) [2], using the official implementation and publicly released training configurations. We build on top of EDM's implementation by adapting the SongUNet and EDMPrecond architectures provided under the VP setting and use the Heun 2nd order solver of EDM sampler for all evaluations in pixel space. Due to the limitation of computing resources, we test LowDiff based on LightningDiT-B/1 and use Euler integrator for all evaluations in latent space. While the core structure of these networks remains unchanged, we introduce minimal modifications to support the generation for different resolutions. Specifically, as mentioned previously, we introduce auxiliary resolution-specific input and output convolutional layers to enable the model to generate images of different resolutions. All training hyperparameters, including noise schedules, loss weighting, optimizer settings, augmentation strategies, and others, are retained to ensure a controlled and fair comparison between our unified model and the original baselines.

We evaluate our model on four benchmark datasets: (1) Unconditional CIFAR-10 Krizhevsky et al. (2009): 50,000 training images of size $32 \times 32$ across 10 classes, without class conditioning. (2) Conditional CIFAR-10 Krizhevsky et al. (2009): The same dataset but with class labels used as additional conditioning inputs. (3) FFHQ64$\times$64 Karras et al. (2019): 70,000 high-quality human face images downsampled to $64 \times 64$ resolution, used in the unconditional setting. (4) ImageNet256$\times$64 Deng et al. (2009): A large-scale dataset consisting of over 1.2 million training images from 1,000 object categories, with all images resized to $256 \times 256$ resolution. It is commonly used in the conditional setting, where class labels serve as conditioning inputs. We use two widely adopted metrics for numerical evaluation: Fréchet Inception Distance (FID) Heusel et al. (2017) which measures the distance between the feature distributions of generated and real images, and Inception Score (IS) Salimans et al. (2016), which evaluates the quality and diversity of generated images based on classifier confidence.

Following the training procedure described in Section 3.4, the number of activated output resolutions is dynamically determined by the target dataset resolution. When training on CIFAR-10 (the resolution is $32 \times 32$), the model includes three output resolution stages: $8 \times 8$, $16 \times 16$ and $32 \times 32$. When training on FFHQ64$\times$64 (the resolution is $64 \times 64$), the model includes four output resolution stages: $8 \times 8$, $16 \times 16$, $32 \times 32$ and $64 \times 64$. When training on ImageNet256$\times$256 (the original latent space is $16 \times 16$), the model includes two output resolution stages: $8 \times 8$ and $16 \times 16$. All models are trained on 8 NVIDIA A40 GPUs.

### 4.2 RESULTS

Table 3: Generation quality and efficiency on CIFAR-10

| Model | No. of Models | Model Size (M) | NFE (res=8) | NFE (res=16) | NFE (res=32) | Uncond. FID ↓ | IS ↑ | Cond. FID ↓ | IS ↑ |
|---|---|---|---|---|---|---|---|---|---|
| EDM | 1 | 55.74 | 0 | 0 | 35 | **2.04** | 9.82 | **1.84** | 9.95 |
| **Ours** | 1 | 55.77 | 18 | 13 | 17 | 2.11 | **9.87** | 1.94 | **10.03** |

Tab. 3, Tab. 4 and Tab. 5 summarize the quantitative comparisons with their corresponding baselines: EDM Karras et al. (2022) and LightningDiT Yao et al. (2025). Compared to their baselines, our models achieve competitive generation quality with comparable FID and better IS on CIFAR10, for both unconditional and class-conditional generation, and give slightly better FID on FFHQ64$\times$64, and much better FID and IS on Imagenet256$\times$256. Moreover, our models significantly reduce the number of truncation evaluations (NFE) at the high resolutions (17 vs. 35, 39 vs. 79, 50 vs. 100).

---

[1] https://github.com/NVlabs/edm

[2] https://github.com/hustvl/LightningDiT

Table 4: Generation quality and efficiency on FFHQ64×64

| Model | No. of Models | Model Size (M) | NFE (res=8) | NFE (res=16) | NFE (res=32) | NFE (res=64) | FID ↓ |
|---|---|---|---|---|---|---|---|
| EDM | 1 | 61.80 | 0 | 0 | 0 | 79 | 2.47 |
| **Ours** | 1 | 61.86 | 22 | 23 | 23 | 39 | **2.43** |

Table 5: Generation quality and efficiency on ImageNet256×256

| Model | No. of Models | Model Size (M) | NFE (res=8) | NFE (res=16) | FID ↓ | IS ↑ |
|---|---|---|---|---|---|---|
| LightningDiT-B/1 | 1 | 130.56 | 0 | 100 | 4.38 | 162.65 |
| LightningDiT-B/1 | 1 | 130.56 | 0 | 50 | 4.72 | 155.68 |
| **Ours** | 1 | 131.86 | 45 | 50 | **4.00** | **195.06** |

To further verify the effectiveness of our method to improve the efficiency of diffusion models, we provide the sampling latency and generation throughput of our models and compare them with other methods, as listed in Tab. 6. To ensure the validity and fairness, the sampling latency for CIFAR10 and FFHQ64×64 is measured on a single NVIDIA GeForce RTX 4090 (24GB) and for ImageNet256×256 is measured on a single NVIDIA A100 (40GB), where each method generates 64 images in a single batch per trial and is averaged over 10 runs. On unconditional CIFAR-10 and class-conditional CIFAR-10, our model achieves a $60.5\%$ speedup in sampling throughput over EDM, reducing latency from 2.22s to 1.37s per batch of 64 images. On FFHQ64×64, our model reduces sampling latency from 12.06s to 7.95s, achieving a $52.8\%$ speedup. On ImageNet256×256, our model reduces sampling latency from 21.87s to 13.23s, achieving a $65.5\%$ speedup. These results demonstrate that our method not only presents strong generative performance but also significantly accelerates the sampling process, offering a satisfied trade-off between efficiency and quality.

The above results demonstrate that our proposed approach which leverages low-resolution generation as a conditioning signal for high-resolution synthesis effectively improves the efficiency of diffusion models. These findings also support our hypothesis that performing denoising at high resolutions throughout the entire generation process is not always necessary, revealing potential redundancy in conventional diffusion pipelines, particularly during the early, noise-dominated stages.

## 4.3 ABLATION STUDIES

**Multi-stage Design.** To explore the effect of low-resolution conditioning across multiple resolution stages, we conducted an ablation study comparing a two-stage model with multi-stage design. On CIFAR-10, we evaluated a two-stage (16-32) model and a three-stage (8-16-32) model, while on FFHQ64×64, we tested a two-stage (32-64) model and a four-stage (8-16-32-64) model. The results are present in the Appendix (Tab. 10 and Tab. 11). The results show that introducing additional low-resolution stages improves sampling efficiency while maintaining image quality. On CIFAR-10, the three-stage model achieves nearly identical FID and IS scores compared to the two-stage baseline but with slightly reduced latency and higher throughput. On FFHQ64×64, the advantage of multi-stage refinement becomes significant where the four-stage model shifts updates to lower resolutions, reducing latency by over one second, increasing throughput by 14%, and achieving a slightly better

Table 6: Sampling Efficiency on Different Datasets (batch size = 64)

| Model | Uncond. CIFAR10 | | Cond. CIFAR10 | | FFHQ-64x64 | | ImageNet-256x256 | |
|---|---|---|---|---|---|---|---|---|
| | Latency (s) | Throughputs (imgs/s) | Latency (s) | Throughputs (imgs/s) | Latency (s) | Throughputs (imgs/s) | Latency (s) | Throughputs (imgs/s) |
| EDM | 2.22 | 28.8 | 2.22 | 28.8 | 12.06 | 5.3 | - | - |
| LightningDiT | - | - | - | - | - | - | 21.87 | 2.9 |
| Ours | **1.37** | **46.7** (↑60.5%) | **1.37** | **46.7** (↑60.5%) | **7.95** | **8.1** (↑52.8%) | **13.23** | **4.8** (↑65.5%) |

FID. These findings confirm that distributing refinement across multiple resolutions is especially beneficial for higher-resolution image generation.

**Unified Model Design.** To better understand the contribution of our unified model design, we conducted an ablation study comparing our two-stage unified model and CDM which includes a separate model for different resolution stage on both CIFAR10 and FFHQ64×64 datasets. The results are summarized in the Appendix (Tab. 12 and Tab. 13). The results show that with the same sampling samples on different resolutions, our two-stage model attains equal or better quality while using fewer parameters and fewer models which requires less memory while sampling. CIFAR-10. Compared to CDM, our two-stage LowDiff reduces parameters from 93.89M to 55.76M on CIFAR-10 and from 117.57M to 61.82M on FFHQ64×64 while improving the generation quality consistently. By unifying all the generation processes into a single model while maintaining a multi-stage generation conditioned on low-resolution images, we eliminate redundancy and achieve better or comparable generation quality. This indicates that model multiplicity is not essential for high performance and that a carefully designed multi-stage model can provide a more practical and resource-efficient alternative to separate cascaded models.

## 5 DISCUSSION AND CONCLUSION

This work introduces a novel multi-resolution framework for building efficient diffusion models. We propose generating low-resolution images as intermediate conditions to guide high-resolution synthesis and design a unified diffusion model that reuses low-resolution sub-networks for both low- and high-resolution generation. This design significantly reduces the overall model size and memory footprint. Our method is trained end-to-end without requiring auxiliary models or additional loss functions, simplifying the training process. Through extensive experiments on several popular datasets, we demonstrate that our approach achieves a superior trade-off between generation quality and efficiency.

**Limitation and Future Work.** While our method focuses on improving the efficiency of image generation pipelines, it also opens several directions for future exploration. One key limitation is that we have not applied our framework to text-to-image generation. Integrating language conditioning within our multi-resolution setup would require a careful redesign of the conditioning mechanism, especially at lower resolutions. Additionally, although our method currently operates in both pixel space and latent space; however, recent advances such as Stable Diffusion have demonstrated the benefits of operating within a compressed latent space (e.g., 64×64). Exploring the use of our framework directly within latent spaces on higher-resolution (e.g., 512×512) could yield further acceleration and reduce memory overhead.

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

APPENDIX

## A   IMPLEMENTATION DETAILS

Our implementation is built upon the official EDM codebase and LightningDiT codebase. We use Pytorch 1.12.1 and CUDA 11.8. We adopt the same configuration for FID calculation as used in EDM and for IS calculation as used in DDGAN Xiao et al. (2021) and the same configuration for FID and IS calculation as used in Guided Diffusion Dhariwal & Nichol (2021).

For training, we preserve all default settings from baseline codebases except for the method of network weight updates and the training duration. Since our model handles images at multiple resolutions within a unified framework, the loss computation requires resolution-specific adjustments. In each training step, we calculate the loss independently for each resolution level. Only the network components actively involved in computing the loss for a specific resolution are updated, allowing resolution-specific parameter update while maintaining computational efficiency. The loss function and corresponding weight remain consistent across all resolutions and follow the original baseline formulation. Our approach involves multiple forward passes within a single training epoch due to resolution-wise processing, the per-epoch training time is longer than the baseline. To maintain a comparable overall training budget, we reduce the total number of training images to 75% of baselines' training settings (i.e., from 200M images to 150M images for EDM and from 64 epochs to 42 epochs for LightningDiT).

## B   EFFECTIVE NFE

The calculation accounts for computational differences between resolution stages by using a latency-based scaling factor derived from per-step generation times. Specifically, we take the ratio of average step latency at low resolution (e.g., 16×16) to target resolution (e.g., 32×32) and apply this factor to convert the low-resolution NFE into equivalent target-resolution NFE. This scaled low-resolution NFE is then added to the actual target-resolution NFE. The precise formula of **latency-based scaling factor** is: $\eta = \frac{\text{latency}_{\text{low}}}{\text{latency}_{\text{high}}}$. The final **Effective NFE** is obtained from $\text{NFE}_{\text{eff}} = \text{ceiling}(\eta \cdot \text{NFE}_{\text{low}}) + \text{NFE}_{\text{high}}$. Although this conversion introduces some approximation error, we mitigate it conservatively by the ceiling function to arrive at the final Effective NFE value. This approach provides a more hardware-aware efficiency comparison than naively summing up NFEs from different resolutions, as it reflects the computational workload difference for different resolutions involved in the cascaded generation.

## C   ADDITIONAL ABLATION STUDIES

To better understand the design choices in our multi-resolution diffusion framework, we conduct ablation studies focusing on three key aspects: (1) the resolution-dependent adjustment of sampling parameters $(\sigma_{\min}, \sigma_{\max})$, (2) the number of sampling steps across resolutions, and (3) the truncation points.

**Resolution-Specific Sampling Parameters.**   Our sampling process consists of two stages: a standard unconditional generation of a $32 \times 32$ low-resolution image, followed by a conditional generation of a $64 \times 64$ high-resolution output. For the first stage, we use the default EDM sampling parameters $(\sigma_{\min}, \sigma_{\max}) = (0.002, 80)$ because this is a standard diffusion process which is the same as EDM. Since the high-resolution generation is conditioned on a structured low-resolution image, we hypothesize that it may not require as wide a noise range, and that using a narrower range could focus denoising on fine-scale details while reducing stochasticity.

To validate this, we conduct an ablation study by independently modifying the sampling parameters at each resolution. As shown in Table 7, narrowing the high-resolution noise range to $(0.01, 50)$ improves FID to 2.46. In contrast, adjusting the low-resolution sampling to $(0.01, 50)$ slightly degrades performance (FID = 2.51), and using narrow ranges at both stages performs worst (FID = 2.65). These results support our hypothesis that standard wide-range sampling is beneficial for

Table 7: Resolution-Specific Sampling Parameters on FFHQ64$\times$64

| $\sigma_{\min}^{32\times32}$ | $\sigma_{\max}^{32\times32}$ | $\sigma_{\min}^{64\times64}$ | $\sigma_{\max}^{64\times64}$ | FID |
|---|---|---|---|---|
| 0.002 | 80 | 0.01 | 50 | **2.46** |
| 0.01 | 50 | 0.01 | 50 | 2.51 |
| 0.002 | 80 | 0.002 | 80 | 2.65 |

Table 8: Truncation Points on Conditional CIFAR-10

| $T_{low}$ | $\widetilde{T}$ | $T_{high}$ | FID |
|---|---|---|---|
| 35 | 17 | 17 | 2.05 |
| 35 | 18 | 17 | 2.01 |
| 35 | 19 | 17 | **1.94** |
| 35 | 20 | 17 | 1.98 |
| 35 | 21 | 17 | 2.03 |

Table 9: Timesteps for Low Resolution on FFHQ64$\times$64

| $T_{low}$ | $\widetilde{T}$ | $T_{high}$ | FID | Latency (s) |
|---|---|---|---|---|
| 59 | 37 | 39 | 2.52 | 8.27 |
| 79 | 49 | 39 | **2.46** | 8.97 |
| 99 | 61 | 39 | 2.46 | 9.80 |

the unconditional stage, while more constrained sampling improves conditional refinement at higher resolutions.

**Truncation Points.** We further investigate the impact of the truncation point $\widetilde{T}$ in our multiscale generation process. The truncation point determines the number of denoising steps performed at the low-resolution stage before transitioning to the high-resolution generation. Intuitively, a proper truncation point ensures that the low-resolution image contains sufficient structure to condition the subsequent high-resolution generation, while avoiding unnecessary computation.

As shown in Table 8, varying $\widetilde{T}$ from 17 to 21 reveals a clear performance trend. FID improves as $\widetilde{T}$ increases from 17 to 19, reaching the best score at $\widetilde{T} = 19$, and then degrades as the value continues to increase. This suggests that moderate truncation allows the model to balance low-resolution generation quality and conditional effectiveness, whereas excessive low-resolution refinement may reduce overall efficiency or lead to suboptimal conditioning.

**Timesteps for Low Resolution.** Building on the truncation-based design, we explore how the total number of denoising steps at the low-resolution stage ($T_{\text{low}}$) affects generation quality and efficiency. Our hypothesis is that due to the presence of truncation at $\widetilde{T}$, increasing the total number of steps beyond this point offers diminishing returns. This is because the effective noise level at the truncation boundary remains similar regardless of the total step count. However, when $T_{\text{low}}$ is too small, the gap between adjacent timesteps becomes larger, potentially leading to instability and quality degradation.

As shown in Table 9, reducing $T_{\text{low}}$ to 59 degrades FID to 2.52. Increasing it to 79 and 99 both improve the result to 2.46, but with increased latency. This confirms that while a minimum number of timesteps is necessary for stable low-resolution generation, excessive denoising steps do not translate into further gains once the truncation level is fixed.

# D  MORE QUANTITATIVE RESULTS

Table 10: Comparison between two-stage model and three-stage model on CIFAR-10

| Model | No. of Models | Model Size (M) | NFE (res=8) | NFE (res=16) | NFE (res=32) | Uncond. FID ↓ | Uncond. IS ↑ | Cond. FID ↓ | Cond. IS ↑ | Latency (s) | Throughputs (imgs/s) |
|---|---|---|---|---|---|---|---|---|---|---|---|
| **Two-stage** | 1 | 55.76 | 0 | 19 | 17 | 2.10 | 9.87 | 1.94 | 10.02 | 1.39 | 45.5 |
| **Three-stage** | 1 | 55.77 | 18 | 13 | 17 | 2.11 | 9.87 | 1.94 | 10.03 | 1.37 | 46.7 |

# E  QUALITATIVE RESULTS

Table 11: Comparison between two-stage model and four-stage model on FFHQ64×64

| Model | No. of Models | Model Size (M) | NFE (res=8) | NFE (res=16) | NFE (res=32) | NFE (res=64) | FID ↓ | Latency (s) | Throughputs (imgs/s) |
|---|---|---|---|---|---|---|---|---|---|
| **Two-stage** | 1 | 61.82 | 0 | 0 | 49 | 39 | 2.46 | 8.97 | 7.1 |
| **Four-stage** | 1 | 61.86 | 22 | 23 | 23 | 39 | 2.43 | 7.95 | 8.1 |

Table 12: Comparison between two-stage LowDiff and CDM benchmark on CIFAR-10

| Model | No. of Models | Model Size (M) | NFE (res=16) | NFE (res=32) | Uncond. FID ↓ | Uncond. IS ↑ | Cond. FID ↓ | Cond. IS ↑ |
|---|---|---|---|---|---|---|---|---|
| CDM | 2 | 93.89 | 19 | 17 | 2.21 | 9.65 | 2.09 | 9.73 |
| **Ours** | 1 | 55.76 | 19 | 17 | **2.10** | **9.87** | **1.94** | **10.02** |

Figure 5 and 6 present generated images for unconditional CIFAR10 and class-conditional CIFAR10 using the trained unified model, which can generate both $16 \times 16$ and $32 \times 32$ images. Figure 7 shows generated images for FFHQ $64 \times 64$ using the trained unified model, which can generate both $32 \times 32$ and $64 \times 64$ images.

Table 13: Comparison between two-stage LowDiff with CDM benchmark on FFHQ64×64

| Model | No. of Models | Model Size (M) | NFE (res=32) | NFE (res=64) | FID ↓ |
|---|---|---|---|---|---|
| CDM | 2 | 117.57 | 49 | 39 | 2.53 |
| **Ours** | 1 | 61.82 | 49 | 39 | **2.46** |

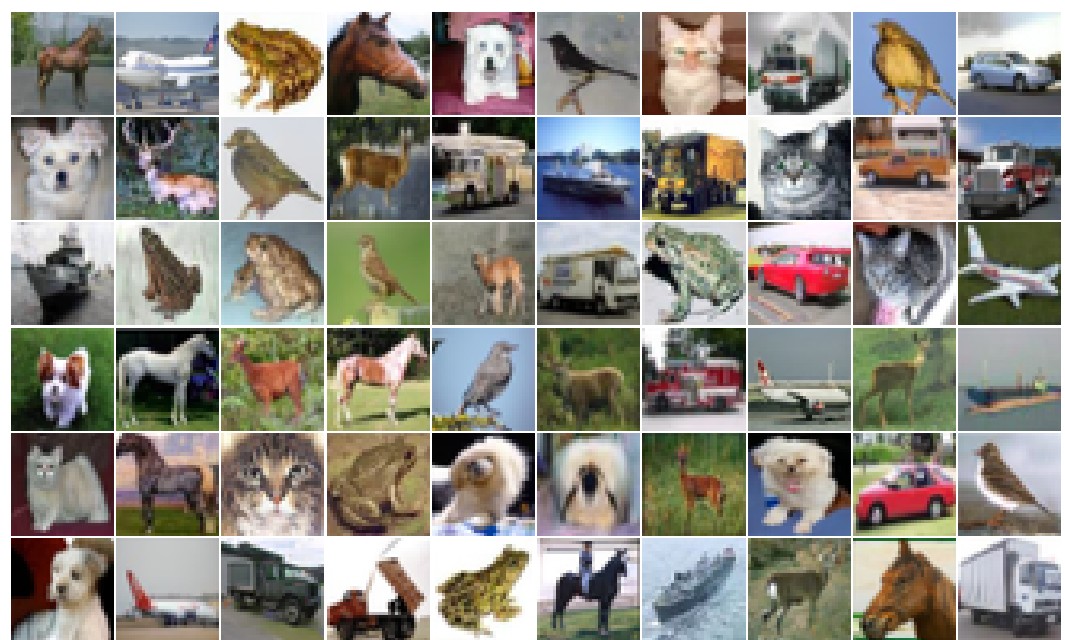

Figure 5: Generated images for unconditional CIFAR10.

Airplane  Bird  Car  Cat  Deer  Dog  Frog  Horse  Ship  Truck

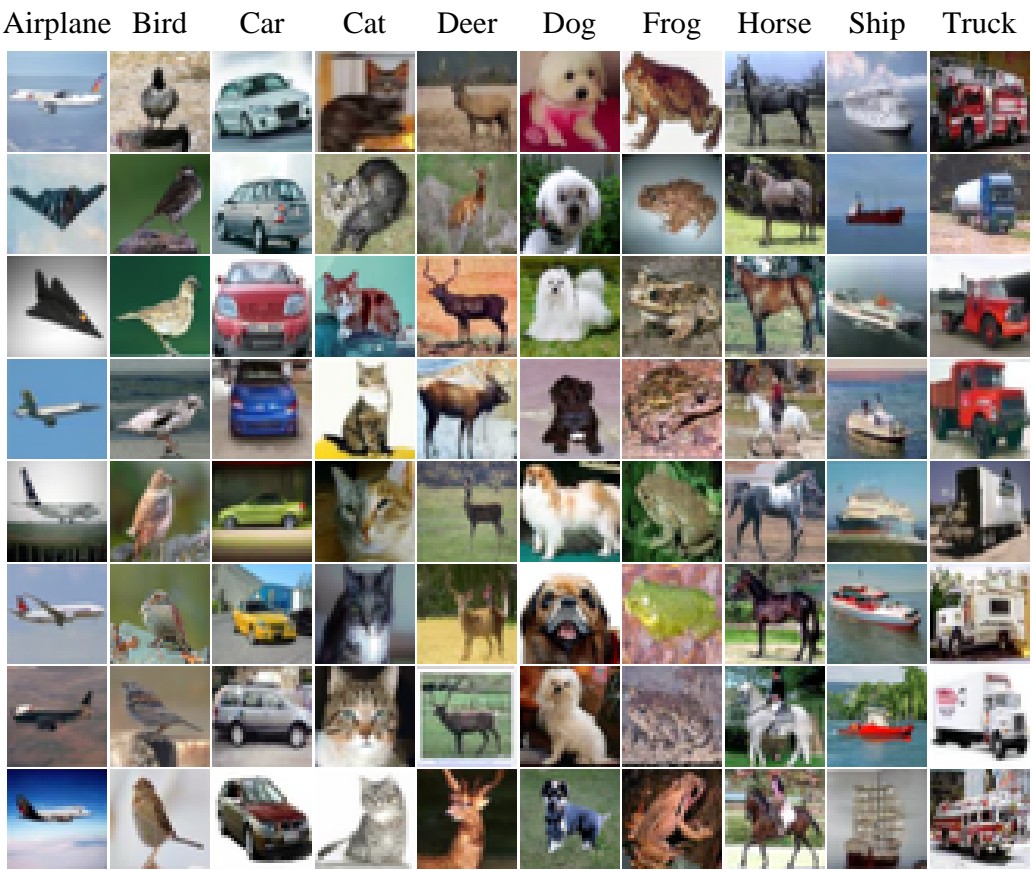

Figure 6: Generated images for class-conditional CIFAR10.

972
973
974
975
976
977
978
979
980
981
982
983
984
985
986
987
988
989
990
991
992
993
994
995
996
997
998
999
1000
1001
1002
1003
1004
1005
1006
1007
1008
1009
1010
1011
1012
1013
1014
1015
1016
1017
1018
1019
1020
1021
1022
1023
1024
1025

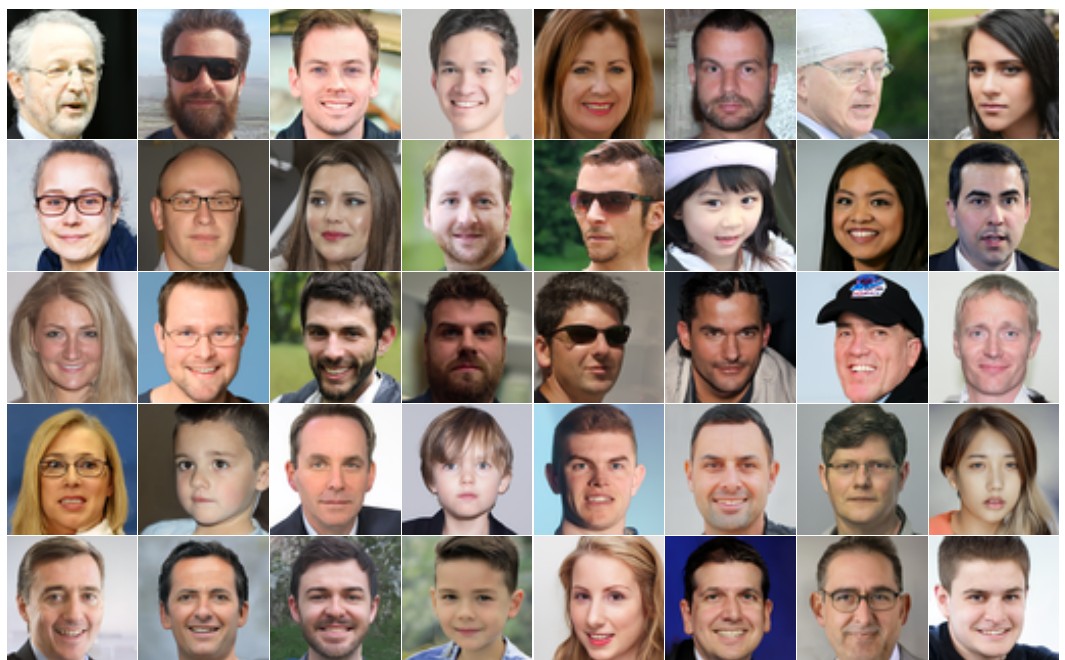

Figure 7: Generated images for FFHQ 64x64.

