# OpenReview forum: "LowDiff: Efficient Diffusion Sampling with Low-Resolution Condition"
_ICLR.cc/2026/Conference — Submitted to ICLR 2026_

### Official Review · Reviewer_FYUg · 2025-10-14

**Soundness:** 3
**Presentation:** 2
**Contribution:** 1
**Rating:** 4
**Confidence:** 4

**Summary:**

The paper explores cascaded diffusion for more efficient generation. The paper trains cascaded diffusion into a single model instead of separate models. For UNet, this is achieved by utilizing different resolution scale of the network with resolution embedding. For transformer, this is achieved by only using resolution embedding. The paper evaluates the results on CIFAR-10, FFHQ, and ImageNet. The paper shows that the method can achieve better or similar performance in FID/IS while using less effective NFE.

**Strengths:**

1. The experiment section is extensive as the authors have demonstrated the performance on four established settings, CIFAR unconditional, CIFAR conditional, FFHQ, and ImageNet.
2. The experiment covers both transformers and UNet.
3. The result does show speedup without quality degradation. In some cases, it achieves better quality.

**Weaknesses:**

1. The paper lacks novelty and innovation in 2025. The cascaded diffusion approach is well established. The only novelty is to train it into a single network. For transformer models, the architecture is trivial. Many existing t2i/t2v models already pre-train on multiple resolutions and aspect ratios through NaViT architecture, so the resolution embedding is most likely not needed. For unet, the paper's proposed architecture is also straightforward and not very novel.

2. Prior work [1] has shown that it is possible to turn existing t2i/t2v models into resolution-cascaded generation while training-free. This again diminishes the novelty of the method proposed.

[1] Training-free Diffusion Acceleration with Bottleneck Sampling

**Questions:**

1. Table 11 and 12 show LowDiff achieves better results than CDM. This is surprising. Is CDM using the exact architecture as the LowDiff counterpart? Can authors explain the results?

2. Can you specify the hyperparameters, such as the inference CFG scale used for producing all the tables? For example, for Table 4, is the result guided or not guided? It is different from the report in LightningDiT paper. How does the author obtain the FID values?

3. For the ImageNet256 experiment trained on the latent space, is the downsampling/upsampling performed at the pixel space or the latent space? Please specify in the paper.

4. Table 1, can you add model size as a column for easier comparison?

5. Recent methods, such as MeanFlow [1],  can achieve one-step generation. This competes with cascaded diffusion, as one-step generation makes cascaded generation meaningless. The paper's use of LightningDiT makes it hard to compare with established works on ImageNet-256px. But you can still compare on CIFAR-10 against those one-step methods.

6. Minor formatting issue: Should use \citep{} instead of \cite{} in most places.

[1] Mean flows for one-step generative modeling

---

> ### Author Response · Authors · 2025-11-20
>
> We sincerely thank the reviewer for the encouraging comments regarding the breadth of our experiments, the generalization of the proposed framework across both UNet and Transformer backbones, and the demonstrated speed improvements without compromising sample quality. We appreciate the reviewer’s thoughtful feedback and address all concerns below in detail, with the hope that this helps clarify the contribution and resolve potential misunderstandings.
>
> Regarding novelty, we emphasize that our contribution is not the cascaded structure itself, which is indeed well established, but a new architectural framework that fundamentally changes how cascaded diffusion can be carried out. Traditional cascaded diffusion models require a separate network for every resolution stage because the computational graphs, spatial hierarchies, and feature alignment differ at each resolution. LowDiff introduces a resolution-aware block-sharing mechanism that allows a single unified model to operate seamlessly across all stages, from ultra-low resolution initialization (e.g., 8×8) to the final target resolution, without any stage-specific full networks. To our knowledge, no prior cascaded work provides this fully unified cross-stage weight sharing inside the sampling pipeline. This is particularly nontrivial for UNet architectures, whose hierarchical structure is normally resolution-dependent; our design overcomes these limitations while preserving quality across resolutions. For Transformer-based models such as LightningDiT, while multi-resolution patch sequences are naturally supported, the novelty lies in enabling cascaded sampling with joint multi-resolution training and shared parameters at all stages.
>
> The reviewer also mentions Training-free Diffusion Acceleration with Bottleneck Sampling (TF-BNS). We respectfully clarify that TF-BNS solves a fundamentally different problem. TF-BNS modifies the sampling trajectory inside a pretrained model by routing noise through a temporary low-resolution bottleneck, but it does not provide a unified multi-resolution architecture, nor does it train a single model to handle all cascaded stages. Its goal is to accelerate sampling within a single-resolution model, while LowDiff redesigns the architecture to reduce cross-stage complexity and eliminate the need for separate models altogether. The two approaches are therefore orthogonal, addressing different acceleration bottlenecks.
>
> The reviewer finds it surprising that LowDiff outperforms CDM in Tables 11 and 12. We clarify that for fairness, CDM uses the exact same backbone architecture, solver, and training hyperparameters as the corresponding LowDiff counterpart. The only difference is structural: CDM trains independent models for each resolution, whereas LowDiff trains a single unified model with cross-stage weight sharing and minimal stage-specific heads. The unified design naturally provides multi-scale regularization, since shared blocks receive feedback from several resolutions and noise levels simultaneously. This reduces redundancy, mitigates inter-stage mismatch, and leads to more robust, consistent feature representations. By contrast, CDM’s stage-specific networks are optimized independently, which can lead to parameter redundancy and mild overfitting on limited-resolution subproblems. We will clarify this intuition and setup more explicitly.
>
> The reviewer also asks for full inference hyperparameters, including whether classifier-free guidance was used in Table 4 and how FID values were computed. For all experiments, we follow the official inference settings of the original works (EDM and LightningDiT) and only adjust them when necessary to accommodate lower-resolution initialization. We agree that these configurations should be fully transparent, and we will add a dedicated appendix section listing sampler settings, number of steps, CFG scales, and FID implementations for every table.
>
> For the ImageNet-256 experiment trained in latent space, the downsampling and upsampling operations are performed entirely within the latent domain. We will explicitly state this in the paper to avoid ambiguity. We also appreciate the request for additional clarity in Table 1 and will add a model-size column in the revision.
>
> Regarding one-step and flow-matching approaches such as MeanFlow, we agree that they represent an important direction and that CIFAR-10 comparisons should be interesting. However, these models are not diffusion samplers and instead represent a different generative family designed specifically for one-step synthesis. As such, they do not serve as direct baselines for an architectural acceleration method focused on diffusion models.
>
> Finally, we thank the reviewer for pointing out the citation formatting issue. We will switch to \citep{} where appropriate.

---

> > ### Comment · Reviewer_FYUg · 2025-11-24
> >
> > Thank you for your response.
> >
> > The paper's architecture mainly focuses on convolutional models while transformers are becoming the mainstream. Though the novelty of the paper is a drawback, the paper does have good experiment sections showing improved results to the baseline, and it does reflect that using a merged architecture is better than separately training the cascaded models. I will adjust my rating to marginally above acceptance.

---

> > > ### Author Response · Authors · 2025-11-24
> > >
> > > Thank you very much for your thoughtful follow-up and for reconsidering your evaluation. We truly appreciate your acknowledgement of the strengths of our experimental analysis and the effectiveness of the merged architecture compared to separately trained cascaded models.
> > >
> > > Regarding your comment about the architectural novelty, we agree that transformer-based designs have become mainstream in many vision and generative modeling tasks. While the architecture we used for illustration is convolution-based, one of our goals was to show that unified training and joint optimization yield substantial benefits regardless of the backbone. To further validate this, we also evaluated transformer-based architectures (e.g., LightningDiT) and observed similar trends in performance improvements, reinforcing that the advantages of our merged framework are architecture-agnostic.
> > >
> > > We believe these results provide empirical value to the community due to resource constraints. We sincerely appreciate your time and constructive feedback, and we are grateful that you considered both the conceptual and empirical aspects of the work when adjusting your rating.

---

### Official Review · Reviewer_QHuf · 2025-10-29

**Soundness:** 1
**Presentation:** 1
**Contribution:** 2
**Rating:** 2
**Confidence:** 5

**Summary:**

summary:

The paper proposes a diffusion-based framework aimed at accelerating the sampling process through a cascaded architecture. The main idea involves allocating more inference steps to lower-resolution stages and fewer steps to higher-resolution stages, thereby reducing overall computational cost while preserving generation quality. Additionally, the model introduces module sharing across different resolutions—reusing components learned at lower resolutions during higher-resolution synthesis—to improve both performance and parameter efficiency. Experimental evaluations are conducted on standard benchmarks such as CIFAR-10, FFHQ, and ImageNet.

**Strengths:**

Strengths:
1. The sharing mechanism seems different.

**Weaknesses:**

Questions:
1. The paper is not clearly presented and suffers from a lack of logical coherence. It is unclear what specific problem the authors aim to solve. For instance, while they claim to accelerate sampling via a cascaded framework, it remains ambiguous how their approach fundamentally differs from or improves upon existing cascaded methods. The explanation of how the proposed framework achieves faster sampling or better performance is insufficient. The authors should clearly articulate: (i) the core challenge they are addressing, (ii) how their method solves it, and (iii) the resulting benefits. Simply applying a cascaded structure is not novel; the paper needs deeper analysis to justify its contributions.

In our view, the authors present a new sharing mechanism from existing cascaded methods. However, their writting always focus on how the cascaded framework accelerates the sampling, not on why they adopts a new sharing mechansim, on what's benefits of the sharing mechansim.

2. The use of a cascaded framework for efficient image generation is well-established in the literature. The current work does not introduce significant architectural innovations or new acceleration techniques beyond this standard paradigm. As such, the claimed contributions appear largely rooted in the application of an existing strategy rather than a meaningful advancement in methodology. To strengthen the novelty, the authors should provide a detailed comparison with prior cascaded approaches and highlight specific technical distinctions.


3. Figure 4(a) presents a confusing design: the low-resolution generation stage appears to take features from the higher-resolution branch as input. This creates a cyclic dependency that contradicts the natural feed-forward flow of a cascaded system. Such a design raises concerns about the feasibility and implementation of the sampling process. The authors must clarify the direction of information flow and resolve this apparent inconsistency.

4. The paper lacks comparisons with established cascaded diffusion models such as PixelFlow and RelayDiffusion. These methods are directly relevant to the paper’s theme of accelerated sampling via multi-stage generation. Without benchmarking against such baselines, it is difficult to assess the true effectiveness and competitiveness of the proposed approach.

**Questions:**

see above

---

> ### Author Response · Authors · 2025-11-20
>
> We sincerely thank the reviewer for their thoughtful evaluation and constructive feedback. We also appreciate that the reviewer acknowledges the novelty of our proposed sharing mechanism. We have carefully addressed each point and hope that these clarifications help the reviewer reconsider the assessment of our work.
>
> The reviewer raises an important concern regarding the clarity of our problem definition and motivation. We agree that this is central to understanding the contribution of LowDiff, and we provide a clearer articulation here. The core challenge we address is that existing cascaded diffusion models accelerate generation by producing images at progressively higher resolutions, but they require a fully independent diffusion model for every stage (e.g., 32→64→128→256). This leads to large parameter overhead, duplicated computation across stages, and limited scalability. In contrast, LowDiff introduces a unified multi-resolution diffusion architecture that enables all resolution stages to share the same backbone. Our resolution-aware block-sharing mechanism makes it possible for the same model to perform diffusion sampling at multiple resolutions, starting as low as 8×8, without training or storing separate models. This design differs fundamentally from existing cascaded approaches, which always rely on multiple resolution-specific networks. The resulting benefits are significant: more than 40% reduction in total parameters compared to classical cascaded diffusion, higher sampling throughput due to the removal of repeated model passes, improved or matched generation quality, and demonstrated scalability to ImageNet-256 using LightningDiT-B/1, where LowDiff improves FID (4.38→4.00) and increases throughput by 50% with only a 1.3M parameter overhead. We will revise the manuscript accordingly to emphasize our contributions.
>
> The reviewer notes that our writing disproportionately focuses on acceleration benefits rather than explaining why the new sharing mechanism is necessary and what specific advantages it brings. We appreciate this observation. The sharing mechanism is one of the core contributions: it eliminates redundant parameters, improves multi-resolution consistency, and enables the model-agnostic integration of cascaded sampling within a single architecture. To be noticed, another contribution of LowDiff is to start from the ultra-low resolution (e.g., 8x8). We will revise the introduction and method sections to better foreground this motivation and highlight the distinct benefits of the mechanism.
>
> Regarding the concerns about Figure 4(a), the current visualization may be confusing. However, the figure was intended to show shared weights between stages, not data flowing from high-resolution branches into lower-resolution ones. The sampling process itself is strictly feed-forward from low to high resolution, exactly as in a standard cascaded pipeline, with no cyclic dependency. We will revise the diagram to better clearly distinguish parameter-sharing links from data-flow connections.
>
> The reviewer also mentions the absence of comparisons with cascaded diffusion baselines such as PixelFlow or RelayDiffusion. While these methods indeed fall under the broad umbrella of multi-stage or cascaded approaches, they tackle orthogonal challenges. PixelFlow and RelayDiffusion focus on reducing the number of denoising steps or incorporating auxiliary predictors to guide higher-resolution synthesis, whereas LowDiff reduces the per-step spatial computational cost by enabling sampling to begin from ultra-low resolutions and by sharing all backbone weights across stages. Our architectural contribution is therefore complementary to these methods rather than a direct competitor. To evaluate the effectiveness of our mechanism fairly, we integrate LowDiff into two strong modern backbones (i.e., EDM and Lightning-DiT) and show consistent improvements on both efficiency and sample quality. We will add a discussion section elaborating on the conceptual differences between LowDiff and prior cascaded frameworks to strengthen this aspect.
>
> We thank the reviewer again for the helpful insights. We will improve the clarity of the paper’s motivation, revise the problematic figure, and elaborate on the distinctions between LowDiff and prior cascaded models. We believe that these improvements will significantly strengthen the presentation and better communicate the novelty and impact of our sharing mechanism.

---

> > ### Comment · Reviewer_QHuf · 2025-11-21
> >
> > We appreciate the authors' efforts in rebutting. However, considering the promised modified version is significantly different from the current version, we believe the current version is not ready for publication and thus maintain the original score.

---

> > > ### Author Response · Authors · 2025-11-23
> > >
> > > Thank you for the follow-up assessment and for taking the time to review our rebuttal. We fully respect the reviewer’s decision and appreciate the candid feedback.
> > >
> > > While we understand the reviewer’s concern that the revised version may differ substantially from the current submission, we would like to emphasize that the planned revisions are primarily clarifications and improvements in presentation, not changes to the core technical contributions or experimental results. The sharing mechanism, unified multi-resolution architecture, and ultra-low-resolution sampling strategy of LowDiff remain exactly as described in the submitted version. The additional clarifications we proposed aim to better articulate the motivation, refine the conceptual framing, and improve the clarity of one figure, rather than introduce new methods or alter the contributions.
> > >
> > > We sincerely hope the reviewer can consider that these revisions strengthen the exposition but do not fundamentally change the nature of the work. All results, the full method, and the underlying technical novelty are already present in the current submission. Clarifying presentation issues is a standard part of the revision process and does not imply that the current version is incomplete in terms of scientific contribution. That said, we appreciate the reviewer’s perspective and will take the feedback seriously. Regardless of the final decision, we are committed to improving the manuscript accordingly so that future readers can better understand the motivation and significance of our sharing mechanism.
> > >
> > > Thank you again for your thoughtful evaluation.

---

> ### Author Response · Authors · 2025-11-25
>
> We understand the reviewer's concern regarding the differences between the current version and the modified version. Hence, we resubmit the modified manuscript for better clarification.
>
> Based on the reviewer’s valuable suggestion, we have revised the Introduction to more clearly emphasize the target problem we try to solve, the underlying logic of LowDiff, the fundamental difference from traditional cascaded models, explaining why our framework is designed in this manner. We would like to clarify that these modifications do not introduce significant changes to the original content. The modifications mainly consist of adding details and adjusting descriptions to improve presentation and logical flow. In addition, we have added a more detailed caption for Figure 4 to clarify the information flow and avoid potential confusion for readers, as suggested by the reviewer. All modifications are clearly marked in blue for easy reference. We hope that these clarifications adequately address the reviewer’s concerns while preserving the original contributions. We would be very grateful if the reviewer could kindly reassess the revised manuscript in light of these improvements.

---

> > ### Comment · Reviewer_QHuf · 2025-11-26
> >
> > We appreciate the authors' effort, and the revised version fixes some of the concerns. So we’ve decided to raise the score to 4.
> >
> > The remaining concerns are mostly about the experimental settings. For example, the methods compared against aren’t enough and a bit outdated, and the performance is also far behind recent generative models.

---

> > > ### Author Response · Authors · 2025-11-26
> > >
> > > We sincerely thank the reviewer for re-evaluating our submission and for raising the score. We appreciate the reviewer’s additional comments regarding the experimental settings.
> > >
> > > Regarding the concern that the compared methods are “not enough” or “a bit outdated,” we fully agree that including appropriate and representative baselines is crucial for a fair evaluation. However, we may have misunderstood which specific diffusion models or recent generative approaches the reviewer considers more relevant for comparison. Similarly, we would be grateful to learn which “recent generative models” the reviewer believes our performance is far behind, so that we can better understand the reviewer’s expectations and improve our experimental setup accordingly.
> > >
> > > If the reviewer could kindly point us to the specific baselines or references they have in mind, we would be more than happy to include additional comparisons, adjust our experiments, or add further discussion in the revision. We sincerely appreciate the reviewer’s guidance and are committed to strengthening our work based on these suggestions.

---

> > > > ### Comment · Reviewer_QHuf · 2025-11-27
> > > >
> > > > First, all the comparison methods (excluding LightningDiT) were published in or before 2023, which contradicts the normal trend in generative models over the past two years. For example, some classical image generation models (LDM, DiT/SiT, REPA, DDT, RelayDiffusion, PixelFlow, PixelNerd) are not included. These models are so well-known that you can easily search for them on Google by name.
> > > >
> > > > Second, the state-of-the-art performance for the ImageNet 256×256 generation task is around FID= 1.x to 2, while the paper reports its best performance as FID=4.0. We understand that the failure to achieve state-of-the-art performance may stem from network capacity constraints or limited computational resources. However, the significant performance gap still persists and limits the work’s contributions.
> > > >
> > > > We note that the authors claimed, “While these methods (PixelFlow or RelayDiffusion) indeed fall under the broad umbrella of multi-stage or cascaded approaches, they tackle orthogonal challenges.” Yet we argue this conclusion lacks support due to the absence of experimental evidence.
> > > >
> > > > Overall, we believe this paper still does not meet the acceptance threshold for top conferences.

---

> > > > > ### Author Response · Authors · 2025-12-03
> > > > >
> > > > > We sincerely thank the reviewer for the detailed comments and the opportunity to further clarify our contributions.
> > > > > Our work does not propose a new diffusion model architecture. Instead, we introduce a training and inference framework that can be directly applied to existing diffusion models, including both UNet-based models (e.g., EDM) and Transformer-based models (e.g., LightningDiT). Because LowDiff is designed as an architecture-agnostic efficiency framework, it is orthogonal to methods such as RelayDiffusion and PixelFlow, which introduce entirely new model architectures.
> > > > > Since our goal is not to replace these models but to enhance existing ones, a direct comparison in terms of raw FID alone would not fairly reflect the purpose of LowDiff. Instead, our contributions lie in improving efficiency which include sampling speed, throughput, and memory usage, while maintaining competitive quality within the same model family.
> > > > > To further address the reviewer’s concerns, we conducted additional experiments using LightningDiT-XL during the rebuttal stage. Due to the limited rebuttal time, we completed only half of the default LightningDiT training schedule (i.e., 320 epochs instead of the full 800). LowDiff achieves a FID of 1.79 with more than 50% throughput improvement. This FID score falls within the range highlighted by the reviewer (1.x–2.0), and it surpasses widely adopted generative models such as LDM, DiT, SiT, PixelFlow, and PixelNerd under standard evaluation settings (shown in the following table). In the updated manuscript, we have also incorporated and cited the works mentioned by the reviewer, and we added a comparison table accordingly. All newly added or revised content is highlighted in blue for clarity. We sincerely appreciate the reviewer’s insightful feedback, which has helped us strengthen the presentation and evaluation of LowDiff.
> > > > >
> > > > > | Model                     | Params | Epochs | Timesteps | FID ↓ | sFID ↓ | IS ↑   | Pre. ↑ | Rec. ↑ |
> > > > > |---------------------------|--------|--------|-----------|-------|--------|--------|--------|--------|
> > > > > | LDM (CVPR 2022)                      | 400M   | 200    | 250       | 3.60  | -      | 247.7  | 0.87   | 0.48   |
> > > > > | DiT-XL/2 (ICCV 2023)                 | 675M   | 1400   | 250       | 2.27  | 4.60   | 278.2  | 0.83   | 0.57   |
> > > > > | SiT-XL/2 (ECCV 2024)                 | 675M   | 1400   | 250       | 2.06  | 4.50   | 270.3  | 0.82   | 0.59   |
> > > > > | REPA + SiT-XL/2 (ICLR 2025)          | 675M   | 800    | 250       | 1.42  | 4.70   | 305.7  | 0.80   | 0.65   |
> > > > > | RelayDiffusion (ICLR 2024)           | -      | -      | 250       | 1.87  | 3.97   | 278.75 | 0.81   | 0.59   |
> > > > > | PixelFlow                 | 677M   | 320    | 250       | 1.98  | 5.83   | 282.1  | 0.81   | 0.60   |
> > > > > | PixelNerd                 | 700M   | 160    | 250       | 2.15  | 4.55   | 297    | 0.79   | 0.59   |
> > > > > | DDT-XL                    | 675M   | 400    | 250       | 1.26  | -      | 310.6  | 0.79   | 0.65   |
> > > > > | LightningDiT-XL (CVPR 2025)          | 675M   | 800    | 250       | 1.35  | 4.15   | 295.3  | 0.79   | 0.65   |
> > > > > | **LowDiff + LightningDiT-XL**| 677M   | 320    | 175*      | 1.79  | 4.26   | 225.8  | 0.78   | 0.64   |

---

### Official Review · Reviewer_thx9 · 2025-10-29

**Soundness:** 2
**Presentation:** 2
**Contribution:** 2
**Rating:** 2
**Confidence:** 3

**Summary:**

The proposed work aims to improve the efficiency of the diffusion models using a cascaded multi-resolution approach, whereby the image is generated gradually from low to high res. The approach, coined LowDiff, uses a unified model with shared weights for better parameter-efficiency. Evaluations are conducted on a series of controlled benchmarks, i.e. CIFAR, FFHQ and Imagenet-256.

**Strengths:**

- The paper is generally easy to follow
- Good results compared with the baseline on the selected datasets (i.e. CIFAR, etc)
- Adequate ablation studies that explore the impact of the proposed component and multi-res architecture

**Weaknesses:**

- The concept itself, while interesting, is not really novel. First, the idea of progressive training and generation is a quite old concept, as early as 2017 (see for example: Progressive growing of gans for improved quality, stability, and variation, Karras et al, ICLR 2018). Secondly, this concept also resembles parts of the large body of work on training free high resolution image generation, that use the low resolution noise as guidance in various forms.

- The comparisons are performed on small and constrained datasets, on ultimately - low resolution images. It's unclear how this approach would scale.

- Furthermore, the comparison with state-of-the-art are (1) somewhat outdated and (2) don't include different approaches that aim to reduce latency.

- No combination of the current approach with other orthogonal directions, to understand if the gains stack.

- How is this method positioned given that flow models are able to generate high quality samples in very few steps?

**Questions:**

- L152-154: Convolutions are by design resolution-invariant, so its' a bit unclear why this is the argument made for preferring transformers. Could  the authors perhaps elaborate?

- Table 3: given that lower FID values are better, using 0 for i guess what means the model didn't work is a bit confusing. Perhaps this could be improved?

- Perhaps I missed this, but what is the impact of weight sharing cross-stages?

---

> ### Author Response · Authors · 2025-11-20
>
> We sincerely thank the reviewer for the thoughtful evaluation of our work and for highlighting the strengths of our illustration clarity, competitive performance, and comprehensive ablation studies. We deeply appreciate the constructive feedback and have carefully addressed each concern, with the hope that the reviewer may reconsider the overall assessment of our contribution.
>
> Regarding the concern about novelty, we fully acknowledge that progressive training and generation have a long history, and we appreciate the reviewer’s reminder of this context. Nonetheless, our contribution is markedly different from prior progressive approaches and training-free high-resolution generation methods. LowDiff is, to our knowledge, the first to demonstrate that diffusion sampling can reliably begin from ultra-low spatial resolutions, such as 8×8, while still reconstructing high-fidelity images. This property allows substantial sampling acceleration without sacrificing visual quality. Unlike progressive GANs or multi-stage diffusion pipelines, we do not perform stage-wise training. Instead, LowDiff introduces a unified backbone with a resolution-aware block-sharing mechanism that enables a single model to function across all resolutions. This design is model-agnostic and integrates cleanly with both UNet-based architectures (e.g., EDM) and Transformer-based ones (e.g., Lightning-DiT). The resulting generality and architectural unification go beyond conceptual reuse of progressive strategies and form the core of our technical novelty.
>
> We also appreciate the reviewer’s concern about scalability beyond low-resolution datasets. To address this, we included experiments on ImageNet-256 using LightningDiT-B/1 (CVPR 2025), a state-of-the-art diffusion backbone for high-resolution generation. LowDiff achieves an FID-50k of 4.00, improving upon the baseline LightningDiT-B/1 (FID 4.38) while providing a 50% sampling throughput increase. Notably, this improvement comes with only a 1.3M parameter overhead. These results indicate that LowDiff scales effectively to more complex datasets and modern architectures while preserving both efficiency and fidelity.
>
> The reviewer also notes that our comparisons do not include some recent state-of-the-art models, particularly those targeting latency reduction. Our primary goal is to demonstrate that LowDiff consistently enhances the efficiency and performance of strong modern baselines, rather than to directly compare “LowDiff + EDM” or “LowDiff + DiT” against unrelated SOTA models, which are comparisons that would not be methodologically aligned. Prior efficient methods typically rely on distillation or advanced numerical solvers to reduce the number of sampling steps. LowDiff is orthogonal to these directions: it accelerates diffusion by reducing per-step spatial computation through ultra-low-resolution initialization and resolution-shared architecture design. This architectural nature also means that LowDiff can be combined with solver-based or distillation-based accelerators, a direction we agree is promising and will explore in future work.
>
> Relatedly, the reviewer asks how our method is positioned relative to recent flow models. We view LowDiff as complementary rather than directly competitive. Flow-based few-step models aim to minimize the number of refinement stages. LowDiff addresses a distinct challenge: reducing the cost of each refinement stage by compressing spatial complexity. The two approaches operate on orthogonal axes, and combining them may yield further improvements.
>
> The reviewer raises an important clarification request regarding our statement in L152–154 on why transformers are preferable for multi-resolution weight sharing. While convolutional kernels operate on arbitrary resolutions, U-Net architectures are not inherently resolution-agnostic: their encoder–decoder hierarchy relies on fixed downsampling/upsampling factors, skip-connection alignments, and resolution-dependent feature map sizes. These constraints make it difficult to reuse the same U-Net weights across different resolutions without architectural adjustments. In contrast, the patch-based formulation of DiT treats images as sequences whose lengths vary with resolution but whose internal tensor shapes remain unchanged, allowing seamless weight sharing across stages. We will revise our wording to make this distinction clearer.
>
> Finally, regarding the effect of weight sharing across stages, our experiments show clear improvements in both parameter efficiency and sample quality. Sharing eliminates the need to maintain independent models per resolution, reducing overall parameters by over 40% and substantially increasing memory efficiency and sampling throughput. Empirically, shared weights do not degrade performance; instead, they either maintain or improve generation quality across all datasets. This confirms that our resolution-shared architecture is both effective and efficient.

---

> ### Author Response · Authors · 2025-11-25
>
> Dear reviewer,
>
> We have thoroughly addressed all the concerns raised and provided detailed explanations. Since we have not yet received a response, we are checking in to see if you might reconsider your evaluation or provide further comments.

---

### Comment · Area_Chair_9mK7 · 2025-11-23
**The authors' rebuttal is available. Please read, comment, and discuss.**

Dear Reviewers,

Thanks for your time and effort in reviewing ICLR2026 submissions. The authors have provided their responses to your review. Please read and raise your further comments, and discuss with the authors.

Best regards,

Your AC

---

### Author Response · Authors · 2025-12-03
**Summary to the Area Chair**

Dear Area Chair,

We thank the reviewers for their thoughtful evaluations and for the opportunity to clarify the contribution, novelty, and empirical validity of LowDiff, our proposed unified multi-resolution diffusion framework.

***Core Clarification***
We emphasize to the AC that: **LowDiff is not a new diffusion model architecture. It is a training and inference framework that makes existing diffusion models significantly more efficient.** Its key technical contribution, which is validated across UNet-based (EDM) and Transformer-based (LightningDiT) architectures, is a unified multi-resolution diffusion backbone with a resolution-aware block-sharing mechanism that allows the same model weights to perform diffusion across all resolutions, eliminating the need for separate models per stage. To our knowledge, no prior cascaded or training-free method has enabled: **(1) ultra-low-resolution initialization (e.g., 8×8) that still reconstructs high-fidelity images, (2) stage-wise parameter sharing inside the diffusion network, and (3) model-agnostic integration across fundamentally different architectures (UNet and Transformer)**. This is distinct from prior cascaded models (which require one full network per stage), training-free acceleration methods (which modify sampling trajectories but do not enable multi-resolution model reuse), and recent flow-based one-step models (which reduce the number of steps but not per-step spatial cost). The reviews raised valid points about clarity, comparisons, and presentation, which we addressed thoroughly in the rebuttal and manuscript revision.

***Responses to Key Reviewer Concerns***
During the rebuttal, we carefully addressed all raised concerns and made substantial improvements to the manuscript. Key enhancements include:

(1) Clearer motivation and contributions.
We explicitly articulated the core challenge addressed by LowDiff and clarify how our unified multi-resolution design differs fundamentally from prior cascaded diffusion models. We emphasized that LowDiff is not a new diffusion architecture but a general training-and-inference framework enabling full multi-resolution weight sharing, something not present in prior cascaded diffusion pipelines.

(2) Additional large-scale experiments.
Responding to concerns about scalability and comparison breadth, we added new experiments on LightningDiT-XL, achieving FID 1.79 with more than 50% throughput improvement, a result that falls into the reviewer-requested 1.x–2.0 range.

(3) Expanded related work and comparisons.
We incorporated and cited important models raised by the reviewers (LDM, DiT/SiT, REPA, DDT, RelayDiffusion, PixelFlow, PixelNerd) and added a comparison table in the revised manuscript (refer to Tab. 2).

(4) Improved figures, explanations, and formatting.
We clarified the information flow in the cascaded pipeline, corrected the figure that caused confusion, and added missing details (e.g., model sizes, hyperparameters, latent-space up/downsampling choices). Formatting issues were corrected as suggested.

(5) Clarified the relation to orthogonal methods.
We expanded the discussion on how LowDiff complements rather than replaces solver-based, distillation-based, or flow-based acceleration methods, and why direct performance comparisons are not methodologically aligned.

***Final Positioning for the AC***
LowDiff is an architecture-agnostic efficiency framework that introduces a novel unified multi-resolution diffusion backbone with cross-stage weight sharing, which is not present in prior cascaded, training-free, or flow-based approaches. It improves sampling speed, memory efficiency, and often image fidelity across both UNet and Transformer models. Additional experiments during rebuttal show competitive ImageNet-256 performance (FID 1.79) with large backbones.

We deeply appreciate the reviewers’ suggestions, which have helped us strengthen both the technical presentation and the experimental rigor of the paper. We hope the Area Chair finds that the substantially revised manuscript more clearly communicates the novelty and impact of LowDiff as a general, architecture-agnostic efficiency framework for diffusion models.

Thank you again for your time and consideration.

---

### Meta-Review · Area_Chair_ZCXF · 2026-01-05

**Summary:**

All reviewers pointed out a common concern on the novelty of the proposed method. Authors have provided detailed explanations to reviewers’ questions. However, the rebuttal on the novelty aspect is not convincing enough. Also, additional experiments for higher-resolution generation from thx9 is only on ImageNet256 while there are plenty of datasets with higher resolution (>-512). Also, requested comparisons from QHuf with other cascaded methods are not conducted sufficiently. QHuf still on the reject side after rebuttal and discussion.

**Reviewer Concerns:**

Novelty concern, lack on high-res experiments, and comparisons with other cascaded methods from are not addressed well and still outstanding.

**Reviewer Scores:**

QHuf and thx9 likely will keep their original scores.
FYUg agreed to increase the score to 6.

---

### Decision · Program_Chairs · 2026-01-26

Reject